# Heterogeneity of Alzheimer's disease identified by neuropsychological test profiling

Truc Tran Thanh Nguyen[1,2¤], Hsun-Hua Lee[3,4,5,6], Li-Kai Huang[4,7], Chaur-Jong Hu[4,7,8,9], Chih-Yang Yeh[10], Wei-Chung Vivian Yang[11], Ming-Chin Lin[1,10,12]*

1 Graduate Institute of Biomedical Informatics, Division of Translational Medicine, College of Medical Science and Technology, Taipei Medical University, Taipei, Taiwan, 2 Memory and Dementia Center, Hospital 30–4, Ho Chi Minh City, Vietnam, 3 Department of Neurology, Taipei Medical University Hospital, Taipei Medical University, Taipei, Taiwan, 4 Department of Neurology, School of Medicine, College of Medicine, Taipei Medical University, Taipei, Taiwan, 5 Dizziness and Balance Disorder Center, Shuang Ho Hospital, Taipei Medical University, New Taipei City, Taiwan, 6 Department of Neurology, Shuang Ho Hospital, Taipei Medical University, New Taipei City, Taiwan, 7 Department of Neurology, Dementia Center, Shuang Ho Hospital, Taipei Medical University, New Taipei City, Taiwan, 8 Graduate Institute of Neural Regenerative Medicine, College of Medical Science and Technology, Taipei Medical University, Taipei, Taiwan, 9 Taipei Neuroscience Institute, Taipei Medical University, Taipei, Taiwan, 10 Graduate Institute of Biomedical Informatics, Division of Biomedical Informatics, College of Medical Science and Technology, Taipei Medical University, Taipei, Taiwan, 11 The PhD Program for Translational Medicine, College of Medical Science and Technology, Taipei Medical University, Taipei, Taiwan, 12 Department of Neurosurgery, Shuang Ho Hospital, Taipei Medical University, New Taipei City, Taiwan

¤ Current address: Taiwan International Graduate Program in Interdisciplinary Neuroscience, National Taiwan University and Academia Sinica, Taipei, Taiwan
* arbiter@tmu.edu.tw

**Data Availability Statement:** The Joint Institutional Review Board of Human Research at the Taipei Medical University has restricted access to the de-identified data set used in this study. Data requests

## Abstract

Alzheimer's disease (AD) is a highly heterogeneous disorder. Untangling this variability could lead to personalized treatments and improve participant recruitment for clinical trials. We investigated the cognitive subgroups by using a data-driven clustering technique in an AD cohort. People with mild–moderate probable AD from Taiwan was included. Neuropsychological test results from the Cognitive Abilities Screening Instrument were clustered using nonnegative matrix factorization. We identified two clusters in 112 patients with predominant deficits in memory (62.5%) and non-memory (37.5%) cognitive domains, respectively. The memory group performed worse in short-term memory and orientation and better in attention than the non-memory group. At baseline, patients in the memory group had worse global cognitive status and dementia severity. Linear mixed effect model did not reveal difference in disease trajectory within 3 years of follow-up between the two clusters. Our results provide insights into the cognitive heterogeneity in probable AD in an Asian population.

## Introduction

Dementia is currently a worldwide epidemic with global prevalence estimates of 50 million people in 2020, in which Alzheimer's disease (AD) accounts for up to 60 to 70% of all patients [1]. The classic presentation of AD was first described in 1906 by Dr. Alois Alzheimer, who

may be sent to Biomedical Technology Building, Shuang-Ho campus, No. 301, Yuantong Road, Zhonghe District, New Taipei City, ohr@tmu.edu.tw.

**Funding:** This research was funded by the Ministry of Science and Technology, Taiwan (grant number 108-2314-B-038-053-MY3) and supported from Taipei Medical University, Taiwan (grant number 108-FRP-02) to the authors. The funders had no role in study design, data collection and analysis, decision to publish, or preparation of the manuscript.

**Competing interests:** The authors have declared that no competing interests exist.

observed a 50-year-old woman with significant memory loss, confusion and aggression [2]. The disorder is now universally recognized by its cardinal symptom–memory impairment that is insidious but progressive and episodic in nature. Recently, "atypical" AD syndromes have also been well-delineated. Specifically, people with posterior cortical atrophy, primary progressive aphasia, and frontal variant initially present with visual impairment, language difficulty, and dysexecutive functions, respectively, instead of the classic fashion of amnesia [3]. Together, these phenotypic variabilities represent the heterogeneity concept of AD. Interestingly, in the 1970s, these subgroups were assumed to belong to different stages and not as genuinely separate entities of AD phenotypes. Subsequent evidence from positron emission tomography (PET) studies and longitudinal follow-ups revealed that they exhibited distinct clinical profiles, brain topographic hypometabolism and disease courses, strengthening the idea of truly unique subtypes [4].

Multiple efforts in disentangling the heterogeneity and complexity of AD have employed data from neuropsychological (NP) assessment, neuroimaging, biomarkers and neuropathology features as the basis for cluster analysis [5–9]. Remarkably, the most consistently found cognitive subtype across studies with different cohorts and clustering techniques is the memory impairment group, versus other groups with more pronounced deficits in other non-memory cognitive domains [5, 7, 8, 10]. For instance, Scheltens et al. employed a data-driven clustering method on neuropsychological measurements and identified two subgroups of AD with discrete cognitive profiles [5, 11]. One subgroup, the "memory" AD, had lower scores in memory-related tests, more severe hippocampal atrophy, slower disease progression and lower risk of mortality, compared to the remaining "non-memory" group that showed impairment in the language or executive/visuospatial domain. This finding strongly indicates that cognitive heterogeneity could predict individuals who are at greater risk of deterioration in dementia.

Importantly, participant samples in most studies investigating heterogeneity in AD are predominantly from the United States and Europe, emphasizing the importance of dedicated funding to dementia research centers to recruit and retain participants, as well as raising questions about the applicability of these findings to patients in other regions of the world. Therefore, this study aims to investigate the cognitive subgroups by using data-driven clustering technique in an Asian population with AD and how this heterogeneity is associated with disease progression.

## Materials and methods

### Patients

Outpatient visits from 2012 to 2020 at the Dementia Center and Department of Neurology of Shuang Ho Hospital, Taipei, Taiwan were identified from medical records. We selected patients diagnosed with probable AD dementia according to the National Institute on Aging and the Alzheimer's Association (NIA-AA) criteria, had available neuropsychological assessment and Mini-Mental Status Examination (MMSE) scores between 16 and 27 at their first visit [12]. We excluded data of people with clinical diagnosis of mild cognitive impairment, mixed and non-AD dementia (i.e. vascular dementia, frontotemporal dementia, dementia with Lewy bodies, normal pressure hydrocephalus, traumatic brain injury), or active psychiatric diseases (schizophrenia, depression with concurrent psychoactive medications usage, suicidality) that could contribute to cognitive deficits. This study was conducted according to the guidelines of the Declaration of Helsinki, and approved by the Joint Institutional Review Board (IRB) of Human Research at the Taipei Medical University (No. N202203146, Form 072/20200317, issue date April 15, 2022). This is a retrospective study of medical records, all

data were fully anonymized before they were accessed, and the IRB committee waived the requirement for informed consent.

## Clinical measurements

Standard dementia evaluation at Shuang Ho Hospital consisted of medical history, physical and detailed neurological examination, neuropsychological (NP) assessment conducted by trained neuropsychologists including MMSE, Cognitive Abilities Screening Instrument (CASI) and Clinical Dementia Rating (CDR) scale, and neuroimaging for most patients. CASI is a NP test battery that can be used to screen for dementia and track disease trajectory [13]. CASI individual items evaluate the following eight cognitive domains: short-term and long-term memory, attention, concentration, orientation, language, visuospatial ability, abstraction and judgment, and are more specifically outlined in the S1 Table. It can be tested within 15–30 minutes in people with dementia and has a total score ranging from 0 to 100, with 0 being the worst and 100 being the best performance. The most common NP evaluation tool in Taiwanese medical institutions, CASI is required by the national health insurance system to provide payment for patients, together with MMSE and CDR [13, 14]. Raw scores of CASI and its items were used in all analyses in this study. CDR was reported as both CDR global score and CDR Sum of Boxes (CDR-SOB), with CDR-SOB found to be more robust in delineating changes within and between stages of dementia severity than CDR global score [15].

Brain MRI data was also available for a subset of patients at baseline examination, including the visual assessment of medial temporal lobe atrophy score (MTA, range 0 to 4) and the Fazekas scale of white matter hyperintensities (WMH, range 0 to 3) [16, 17]. For the atrophy scores, we computed the mean values of right and left hemispheres, while for the WMH scale we reported the deep white matter component score as it was more relevant to neurodegenerative diseases than the periventricular white matter component [16]. Of note, while the use of neuroimaging software, such as FreeSurfer [18], to perform cortical parcellation and subcortical segmentation would provide a richer dataset in this study, we have opted to include the MTA and Fazekas measures, which are visual rating scales that can easily be applied in clinical settings.

## Clustering analysis

Nonnegative matrix factorization (NMF) was applied to identify cognitive subtypes of people with AD, using the R package *NMF* version 0.24 [19]. NMF is an unsupervised machine learning algorithm that decomposes (or factorizes) a matrix *V* into two matrices *W* x *H*, with all three matrices containing no negative entries [20, 21]. Hence, it is a dimensionality reduction technique with intrinsic clustering property [22]. In the context of this study, "matrix" V represents the *n* x *p* table of NP scores of all patients, with each row *n* equivalent to one of seven cognitive domains (we omitted the item "Orientation" in the clustering analysis as it was not one of the major cognitive domains being assessed in two primary AD diagnostic criteria [12, 23]) and each column *p* equivalent to one patient (Fig 1). Accordingly, *W* is the "meta"-cognition matrix with size *n* x *r*, and *H* is the "meta"-cognition performance profile with size *r* x *p*. Of note, *r* is the factorization rank, i.e. the number of clusters. With no *a priori* assumptions of the cognitive subgroups, as is the data-driven nature of this study, it is important to determine the optimal number of clusters, *r*, based on the NMF algorithm. We calculated the cophenetic correlation coefficients, the most common approach to choose *r*, of *r* ranging from 2 to 6 and performed 50 iterations to obtain a robust estimate of *r* [19]. The default "brunet" NMF algorithm was implemented to identify the decomposed matrices *W* and *H* based on their Kullback-Leibler divergence distance from the original matrix *V* [24].

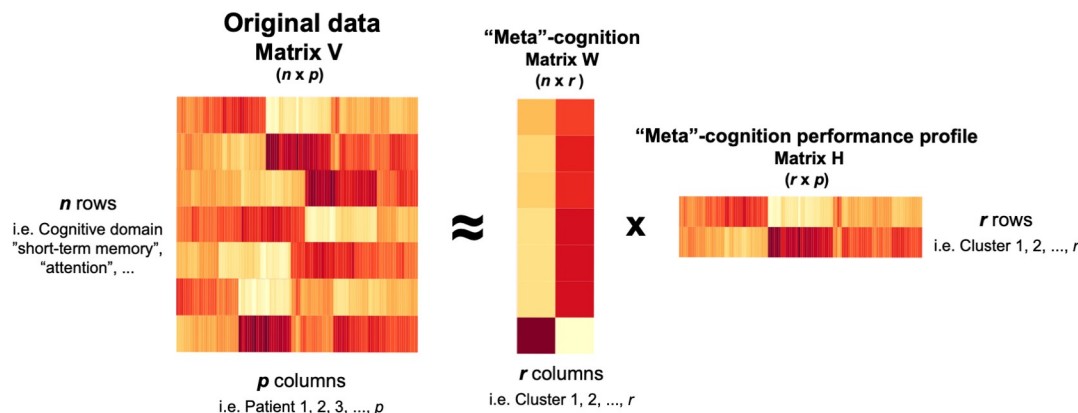

**Fig 1. Schematic overview of the nonnegative matrix factorization (NMF) algorithm.**

### Statistical analysis

For the complete patient dataset and each cognitive clusters, we reported the demographic characteristics (age, gender, education in years, duration of onset, whether patients had multiple tests available from annual visits) and clinical measurements (NP test results, dementia severity stage, MRI visual ratings of brain atrophy and white matter lesions). Comparisons of these characteristics between the cognitive clusters were made using $t$ test, Wilcoxon signed-rank test or $\chi^2$ test where applicable.

As the majority of patients in our study had more than one annual visit and NP evaluation, we further investigate the association between cognitive cluster membership and longitudinal change in NP tests results and dementia severity stage with linear mixed-effect model, using the R package *lme4* version 1.1–29 [25]. Outcome variables were MMSE, CASI, CDR global score and CDR-SOB. Predictors were cognitive cluster, time, and the interaction between cluster and time. We implemented a random intercept and random slope model to account for variabilities in NP performances of each patient. Analysis was limited to within 3 years of follow-up visits to avoid possible effect of dropout (i.e. people with more severe diseases are less likely to undergo NP evaluation) and an imbalanced dataset unfavorable for the linear mixed-effect model [26].

Of note, there was virtually no missing data in the NP test and dementia severity grading scores, as these measurements were mandatory for patients with dementia to receive treatments covered by insurance and the coverage rate of national health insurance in Taiwan is approximately 99% [27]. On the other hand, the duration of onset (the time from when symptoms first presented to patients' first hospital visit) was not available for all patients, due to either the patients or family members not providing this information or it was not reported in the medical records by physicians. Similarly, some patients received brain CT scans instead of MRI or had no neuroimaging, leading to missing ratings of brain atrophy and white matter lesions. No patients were excluded because of missing values, which were handled via pairwise deletion in respective analyses.

All computations were performed with R version 4.2.0 and RStudio version 2022.02.2+485 on Macbook [28]. Statistically significant $p$ values were set at $< 0.05$.

## Results

### Patient characteristics

A total of 112 patients with probable AD were included (Table 1). At baseline, they were 77 ± 7.9 years old and 60% of them were female. Accounting for their duration of onset of 1.3 ± 0.96 years (which were available for 65% of the patients), the age at which they first

**Table 1. Patient characteristics.**

| Characteristics | Total | | Memory | | Non-memory | | p value |
|---|---|---|---|---|---|---|---|
| | n = 112 | | n = 70 (62.5%) | | n = 42 (37.5%) | | |
| *Demographic* | | | | | | | |
| Age | 77 ± 7.9 | (54–99) | 77.3 ± 7.5 | (54–92) | 76.3 ± 8.5 | (58–99) | 0.45 |
| Age at onset[a] | 75.4 ± 8.3 | (50–98) | 75.1 ± 7.8 | (50–91) | 75.9 ± 9.2 | (57–98) | 0.71 |
| Gender (female %) | 68 | (60%) | 38 | (54%) | 30 | (71%) | 0.1 |
| Education (years) | 7.9 ± 4.06 | (0–18) | 8.4 ± 4.1 | (0–18) | 7 ± 3.7 | (0–16) | 0.13 |
| Duration of onset (years)[a] | 1.3 ± 0.96 | (0.5–4) | 1.3 ± 1 | (0.5–4) | 1.28 ± 0.89 | (0.5–3) | 0.85 |
| Multiple tests available (yes %) | 97 | (86%) | 58 | (82%) | 39 | (92%) | 0.22 |
| Duration of follow-up (years) | 2.8 ± 1.8 | (0–8) | 3.4 ± 1.8 | (0–8) | 2.5 ± 1.7 | (0–6) | ***0.009*** |
| *Cognitive measurements* | | | | | | | |
| MMSE | 21.1 ± 3.1 | (16–27) | 20.2 ± 2.5 | (16–26) | 22.5 ± 3.4 | (16–27) | ***0.0001*** |
| CASI | 70.1 ± 9.8 | (40–92) | 67.8 ± 8.8 | (40–86) | 73.9 ± 10.4 | (51–92) | ***0.002*** |
| Short-term memory | 5.3 ± 2.8 | (1–11) | 3.4 ± 1.4 | (1–7) | 8.3 ± 1.6 | (5–11) | ***< 2.2e-16*** |
| Long-term memory | 9.5 ± 1.04 | (5–10) | 9.5 ± 1.06 | (5–10) | 9.3 ± 1 | (7–10) | 0.08 |
| Attention | 5.3 ± 2.8 | (2–8) | 6.3 ± 1.2 | (3–8) | 5.6 ± 1.5 | (2–8) | ***0.01*** |
| Concentration | 6.2 ± 2.2 | (0–10) | 6.3 ± 2.2 | (1–10) | 6.02 ± 2.2 | (0–10) | 0.51 |
| Orientation | 12.2 ± 3.8 | (4–18) | 11.1 ± 3.8 | (4–18) | 14.1 ± 3.2 | (7–18) | ***9.03e-05*** |
| Abstraction–Judgment | 8 ± 1.9 | (3–12) | 8.01 ± 1.9 | (4–12) | 7.9 ± 2.07 | (3–12) | 0.99 |
| Language | 14.5 ± 2.8 | (6–20) | 14.8 ± 2.7 | (7–20) | 14.1 ± 2.9 | (6–20) | 0.26 |
| Visuospatial | 8.1 ± 2.4 | (0–10) | 8.1 ± 2.5 | (0–10) | 8.2 ± 2.4 | (0–10) | 0.91 |
| CDR global score | 0.6 ± 0.2 | (0–2) | 0.6 ± 0.2 | (0.5–2) | 0.5 ± 0.2 | (0–1) | 0.15 |
| CDR-SOB | 2.4 ± 1.9 | (0–11) | 2.8 ± 2.09 | (0.5–11) | 1.6 ± 1.3 | (0–5) | ***0.0006*** |
| *Neuroimaging*[b] | | | | | | | |
| MTA | 1.18 ± 0.91 | (0–3.5) | 1.25 ± 0.99 | (0–3.5) | 1.07 ± 0.78 | (0–2.5) | 0.49 |
| WMH | 1.58 ± 0.91 | (0–3) | 1.54 ± 0.9 | (0–3) | 1.65 ± 0.93 | (0–3) | 0.58 |

*Abbreviations*: CASI = Cognitive Abilities Screening Instrument, CDR = Clinical Dementia Rating, CDR-SOB = Clinical Dementia Rating Sum of Boxes, MMSE = Mini Mental State Exam, MTA = medial temporal lobe atrophy score, WMH = white matter hyperintensities (Fazekas scale).

Note: Data are presented as mean ± standard deviation with range (min—max). P values were computed from *t* test, Wilcoxon signed-rank test or $\chi^2$ test where applicable.

[a] Missing in 35.7% (40/112) of the cohort, 34% (24/70) of the memory group and 38% (16/42) of the non-memory group.

[b] Missing in 23.2% (26/112) of the cohort, 28.5% (20/70) of the memory, 16.6% (7/42) of the non-memory group.

presented symptoms was 75 ± 8 years. Nearly all patients (86%) had multiple ($\geq$ 1) visits to the Dementia Center or Department of Neurology, with an average of 2.8 ± 1.8 years of follow-up. Patients had mild to moderate AD dementia, with mean MMSE, CASI, CDR global and CDR-SOB scores being 21.1, 70.1, 0.6 and 2.4 respectively. As expected, they performed relatively well in NP assessments of long-term memory (mean score 9.5, the highest possible score is 10), language (mean 14.5, highest possible 20), and visuospatial ability (mean 8.1, highest possible 10), which are largely preserved cognitive domains until the late stages of AD [29]. MRI scans were available in 76% of the cohort (86 of 112 patients), with mean MTA and WMH scores of 1.18 and 1.58, respectively.

## NMF-based clustering analysis revealed two subgroups with prominent deficits in memory versus non-memory cognitive domains

We first identified the optimal number of cognitive clusters, *r*, by comparing the cophenetic correlation coefficients of *r* ranging from 2 to 6. S1 Fig revealed that two was the most robust

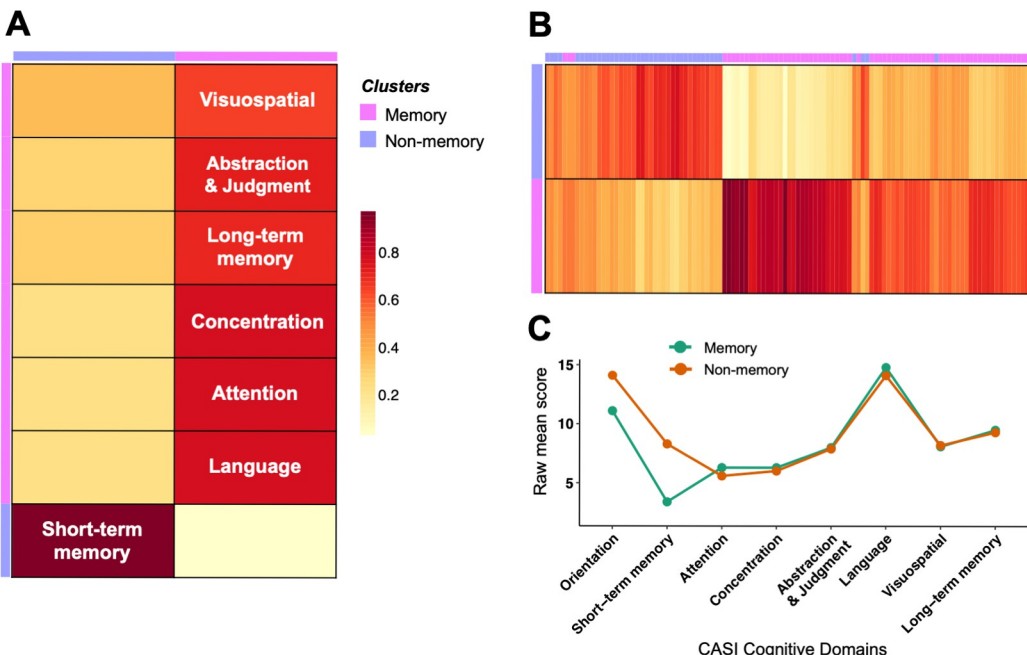

**Fig 2. Results of NMF clustering analysis.** (A) Heatmap generated from the "meta"-cognition matrix *W* with rows representing 7 cognitive items of CASI and columns representing two cognitive clusters. The cognitive items are colored using a red–yellow scale. The closer to the red hue, the higher the item scores are (better cognitive performance) and vice versa. (B) Heatmap from the "meta"-cognition performance profiles matrix *H* with rows representing two cognitive clusters and columns representing each patient. (C) Comparison between CASI items of two cognitive clusters, memory and non-memory. CASI = Cognitive Abilities Screening Instrument.

number of clusters, with the highest cophenetic coefficient at approximately 0.95. Next, results of the NMF algorithm with *r* = 2 cognitive subtypes are shown in Fig 2. The first cognitive cluster included CASI items long-term memory and other non-memory domains, while the remaining cluster only included short-term memory (Fig 2A). Fig 2B shows the "meta"-cognition profiles matrix of the patients as a heatmap, with each column corresponding to one patient. Columns were arranged by cluster membership, i.e. either one of the two clusters that the patient belongs to, as illustrated by the uppermost thin row (pink and blue colors). Subsequent comparison of the NP scores of the two clusters (Fig 2C and Table 1) confirmed that the patients belonging to the first cluster had significantly lower scores in short-term memory and orientation and higher scores in attention, while scores in the remaining categories were similar between the two subgroups. Hence, we identified the first cluster as the "memory" cluster for their primary deficit in memory, accordingly the second cluster was named "non-memory".

We next examined the demographic and clinical characteristics of the two clusters (Table 1). Importantly, memory and non-memory patients were similar in age, gender, years of education, and duration of onset, demonstrating that these possible confounding effects were not driving the heterogeneity found from cognitive clustering [9]. Both groups were also comparable in levels of medial temporal lobe atrophy and cerebrovascular burden. In terms of cognition-related measurements, the memory group had lower MMSE and CASI and higher CDR-SOB scores, indicating that they had worse global cognitive status and dementia severity at baseline.

## Comparable rate of dementia progression between the two subgroups

Next, we investigated the disease progression of patients in two cognitive clusters. The trajectory of MMSE, CASI, CDR global and CDR-SOB raw scores during 5 years of follow-up is illustrated in Fig 3. Patients in both groups showed considerable variability in changes of CASI, MMSE and CDR scores, with some progressing to more severe stages within a few years while others appeared to be relatively stable.

We employed linear mixed effect model on a subset of patients with available follow-up results (n = 97 [86% of the original dataset], memory group n = 58 [60%], non-memory group n = 39 [40%]). We only included data within three years of follow-up as from the fourth year on the number of participants with available data dropped substantially (n = 43 from 112), which could induce the linear mixed effect model algorithm to fail to converge, i.e. it would not reach a stable solution with maximum likelihood. It was revealed that within three years of follow-up, patients in both groups significantly worsened over time (Fig 4 and Table 2). Specifically, patients in the memory group were predicted to have annual MMSE decreased by 1.1 ± 0.2 points, CASI decreased by 3.8 ± 0.7 points, and CDR-SOB increased by 0.8 ± 0.1 points. In the non-memory group, patients were estimated to have MMSE decreased by 1.2 ± 0.3 points, CASI decreased by 3.3 ± 0.8 points, and CDR-SOB increased by 0.7 ± 0.1 points per year. However, the interaction between follow-up time and cluster membership was not statistically significant for all cognitive measurements (p values from 0.6 to 0.99), suggesting that both memory and non-memory groups had similar rates of progression.

## Consistency of clustering analysis over time

Finally, we performed NMF clustering analysis on patient's NP test scores at year 1, 2, and 3 after the baseline visit. Remarkably, Fig 5 shows that patients at each year for the first 3 years

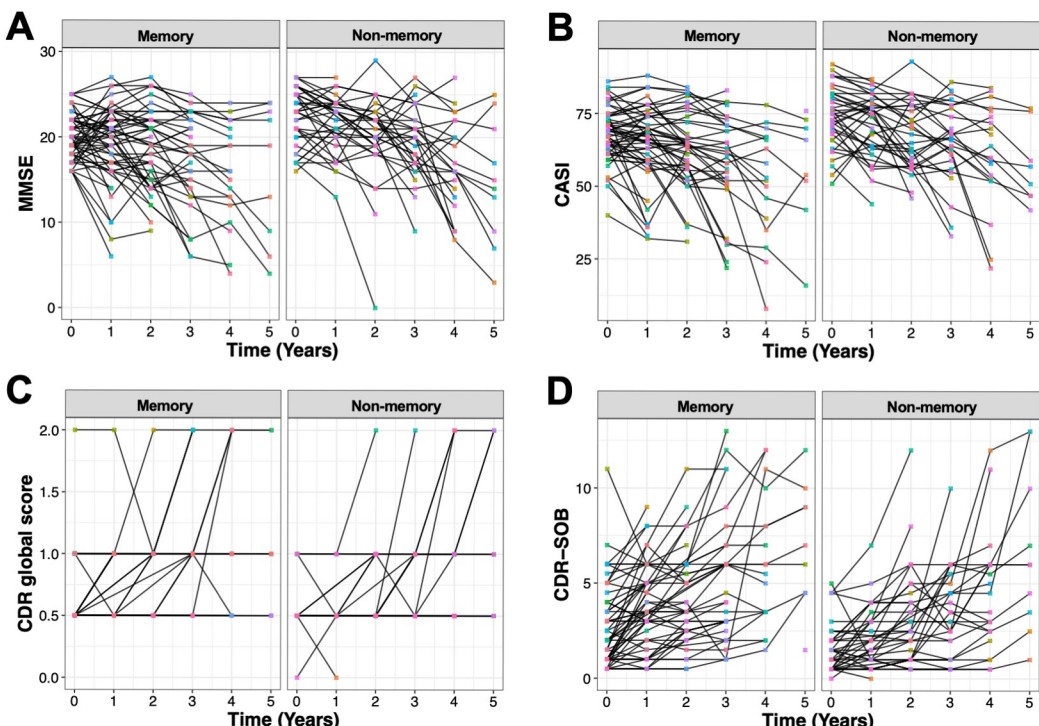

**Fig 3. Longitudinal trajectory of cognitive measurements during 5 years of follow-up (raw scores).** (A) MMSE, (B) CASI, (C) CDR global score, (D) CDR-SOB.

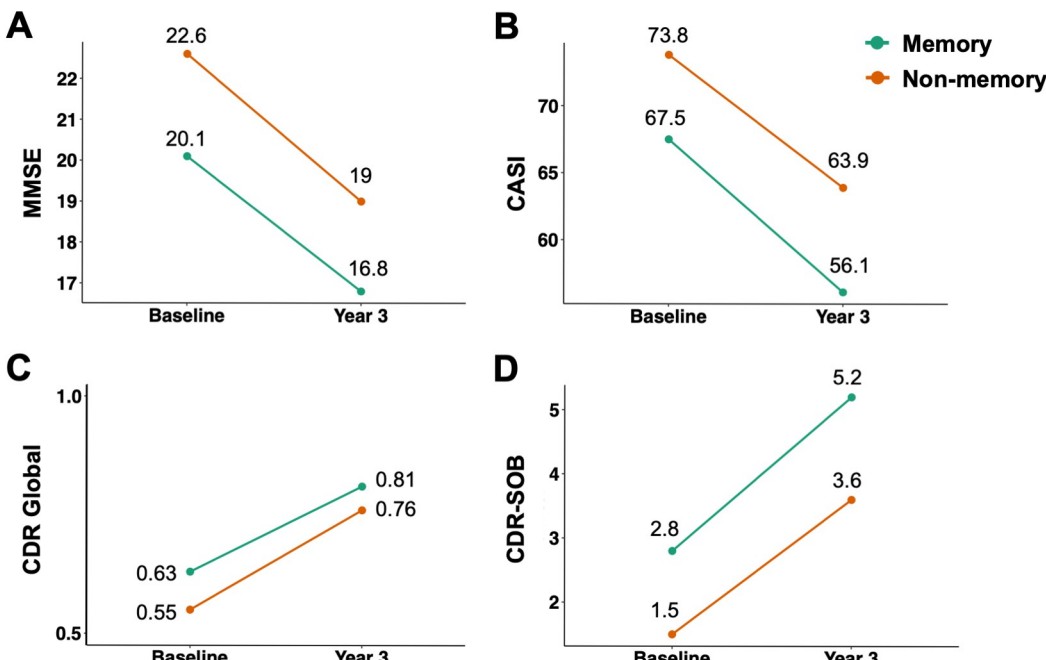

**Fig 4. Predicted change in cognitive measurements after 3-year follow-up.** (A) MMSE, (B) CASI, (C) CDR global score, and (D) CDR-SOB.

are clustered similarly to their baseline cognitive performance, i.e. one group with primary deficits in short-term memory and the other in the remaining domains, emphasizing the consistency of the 2-cluster solution over time.

We further analyzed the stability of this clustering results over the year, as in whether patients consistently belong to the same group or shift their cognitive membership. As shown in Fig 6, the rates of stability, calculated as the sum of patients with similar cluster membership as the previous year divided by the total visits of that year, ranged from 71% to 85% over 3 years, indicating that the majority of patients displayed a steady profile of cognition with regard to memory versus non-memory.

## Discussion

### Emphasizing the 2-cluster solution in cognitively heterogeneous AD

In this study, we implemented NMF, a data-driven clustering technique, to investigate the cognitive heterogeneity in people with mild to moderate AD and found two subgroups showing

**Table 2. Longitudinal analysis of cognitive measurements.**

|  | Baseline | | | Predicted annual change | | |
|---|---|---|---|---|---|---|
|  | Memory | Non-memory | *p* value | Memory | Non-memory | *p* value |
| MMSE | 20.1 ± 2.6 | 22.6 ± 3.3 | ***0.0001*** | −1.1 ± 0.2 | −1.2 ± 0.3 | 0.6 |
| CASI | 67.5 ± 8.9 | 73.8 ± 10.1 | ***0.002*** | −3.8 ± 0.7 | −3.3 ± 0.8 | 0.7 |
| CDR global | 0.63 ± 0.2 | 0.55 ± 0.2 | 0.1 | 0.06 ± 0.02 | 0.07 ± 0.02 | 0.99 |
| CDR-SOB | 2.8 ± 2.1 | 1.5 ± 1.3 | ***0.002*** | 0.8 ± 0.1 | 0.7 ± 0.1 | 0.6 |

*Abbreviations*: CASI = Cognitive Abilities Screening Instrument, CDR = Clinical Dementia Rating, CDR-SOB = Clinical Dementia Rating Sum of Boxes, MMSE = Mini Mental State Exam.

Note: For baseline scores, data are presented as "mean ± standard deviation" and *p* values computed from *t* test or Wilcoxon signed-rank test. For predicted annual change, data are presented as "estimate ± standard error" and *p* values were computed from linear mixed effect model analysis.

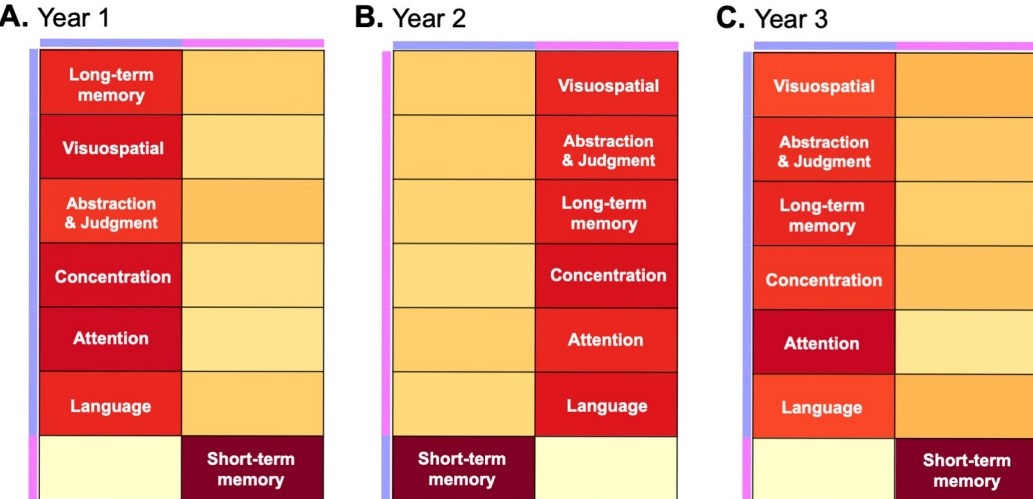

**Fig 5. Heatmaps generated from the "meta"-cognition matrix W at Year 1, 2, and 3 of patients' NP test results.**

prominent deficits in memory (62.5%) and non-memory (37.5%) cognitive domains. The higher number of patients in the memory subgroup was expected as people with dementia most commonly seek medical care due to memory complaints. This result is largely in line with previous studies that identified two cognitive subgroups in AD, with the memory or typical group comprising 48% to nearly 80% of the sample [5, 7]. Further scrutiny of the NP profile revealed that non-memory patients performed better on two cognitive domains (namely, short-term memory and orientation) and worse on attention compared to the memory group; while scores on concentration, abstraction–judgment, visuospatial ability and long-term memory were similar across both groups. Comparably, Qiu *et al.* reported that AD atypicality was

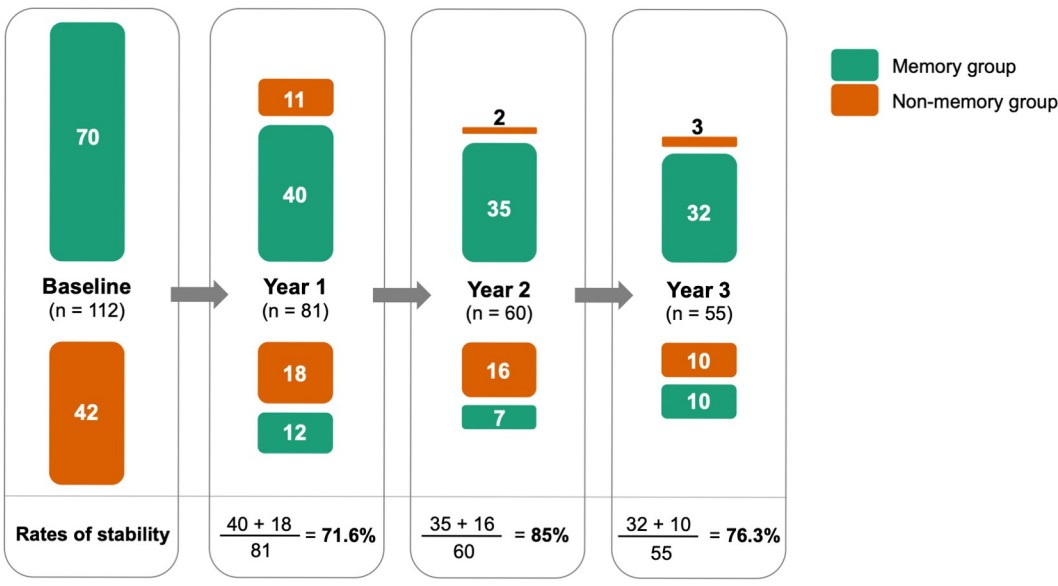

**Fig 6. Stability of the 2-cluster solution at Year 1, 2, and 3.** Columns were scaled according to the patient count in the memory (green) or non-memory (orange) groups.

associated with better episodic memory and worse attention/executive function, and that typical and atypical patients had similar scores in language and visuospatial skills [7].

One particular finding of the study was that patients assigned to the memory group had more severe dementia (lower MMSE and CASI, higher CDR-SOB scores) than the non-memory group at baseline. This could imply that the variance identified from subtype analysis was actually driven by different disease stages, thus not true cognitive heterogeneity per se [9]. On the other hand, one could argue that temporal and phenotypic heterogeneity are unavoidably intertwined in diseases and develop algorithms that take into account both simultaneously, as was performed in Young et al. [30]. In our study, we believe that disease stage did not substantially contribute to the identification of cognitive subgroups, because age at onset and duration of illness did not differ significantly between the two groups. While we could have validated the cognitive clusters following stratification for disease severity, i.e. split the cohort into two groups using the median MMSE scores at baseline and repeated the cluster analysis on each group to see if the results would be similar [5], we elected not to because splitting the original cohort of 112 participants further would likely result in too small sample size.

Two important results of our study diverge from previous works and merit serious consideration. Firstly, we observed no group differences in age, gender, duration of onset, and neuro-imaging variables, in contrast to three studies reporting that patients in the non-memory cluster were younger, more often male and had less severe hippocampal atrophy. [5–7]. It has been suggested that subgroups identified from unsupervised clustering techniques should be interpreted with caution, as they could be driven by variances unrelated to disease expression patterns, like age, gender and disease stages [9]. In this respect, it could be argued that our 2-cluster finding is less likely to have been confounded by these variables. Furthermore, it is possible that MRI visual rating scores, while can be routinely acquired in the clinical setting, could not capture brain regions as fine-grained as specialized neuroimaging software suites, e.g. FreeSurfer. Secondly, we discovered that cluster membership, memory versus non-memory, did not contribute to differences in disease progression after 3 years of follow-up. Qiu *et al.* performed cluster analysis on a sample of more than 4,000 patients from the U.S.'s National Alzheimer's Coordinating Center and revealed that atypicality was significantly related to less severe dementia and slower cognitive decline [7]. Conversely, Scheltens *et al.* included a total of 1,066 AD patients from the Netherlands, German and two U.S. sites (Alzheimer's Disease Neuroimaging Initiative–ADNI and University of California San Francisco–UCSF) and showed that the non-memory group had worse MMSE scores and faster disease progression, but only in the UCSF cohort. Apparently, the relationship between memory versus non-memory impairment with global dementia and longitudinal rate of cognitive decline remains open to question.

Overall, our findings highlighted the existence of two distinct subgroups with cognitive deficits in memory and non-memory domains. Previous studies striving to identify clusters of cognitive impairments in AD have reported from two to as many as thirteen clusters [5–7, 31–36]. Approaches varied mostly in terms of patient inclusion criteria, types of NP tests and clustering methods. For example, studies either included patients at all stages of AD [32, 33, 36] or only those in the mild and moderate stages, using MMSE cut-off values at 10 [6], 14 [34], 15 [5] or 16 [7] as one of the inclusion criteria to avoid the impact of floor effects on clustering results. In addition, aside from the typical "memory" subgroup that was consistently found across studies, researchers have also identified other subgroups characterized by cognitive impairment other than memory or those at different levels of cognitive severity. Previous work by Cappa et al., which employed one of the most comprehensive NP battery, found four hierarchical clustering-derived subtypes, namely the memory, visuospatial/perceptual, perceptual/calculation, and language from a cohort of patients with AD and posterior cortical atrophy

[32]. Similarly, Zangrossi et al. identified four AD groups– typical, mild, visuospatial, and non-amnestic–that were partly explained by differences in sex and global cognitive functioning [37]. Scheltens et al. employed latent class analysis and identified eight AD cognitive clusters characterized by discrete neuropsychological performance and disease severity [6]. Evidently, the more subgroups recognized by distinct clustering algorithms, the more detailed and fine-grained these subtypes are revealed [9]. However, we believe that while not a single number of clusters is yet considered to be optimal and clinically meaningful, there is a trade-off between the degree of clustering resolution and the ability to interpret and apply this heterogeneity in the clinical settings.

## Heterogeneity incorporating multi-dimensional data and its underlying mechanisms

The present study focused on investigating the heterogeneity in patients with AD using results from NP assessment. We employed CASI, a battery of NP test commonly used in medical institutions across Taiwan and multi-center cohort studies worldwide (e.g. Multi-Ethnic Study of Atherosclerosis–MESA [38, 39], Adult Changes in Thought–ACT study [40, 41], and Shiga Epidemiological Study of Subclinical Atherosclerosis–SESSA [42]). In comparison with NP measures used elsewhere, the CASI consists of a more comprehensive set of cognitive domains (with the addition of item "Abstraction and Judgment", which is rarely assessed formally in cognitive evaluations [13]) and has been specifically modified from the original English version to accommodate Chinese elders of limited educational backgrounds (outlined in S1 Table) [43]. However, several items of the CASI could be regarded as more limited in scope, especially the "Short-term Memory" item, which only required participants to recall three words and five objects, compared to the more exhaustive Rey Auditory Verbal Learning Test and Logical Memory test used in the multisite ADNI study, which assess the ability to recall 15 words and a short story, respectively [44–46].

Certainly, NP pattern is not the only modality in dementia that has been leveraged to disentangle its heterogeneity. Extensive works have applied clustering methods on neuroimaging (structural/functional/diffusion tensor MRI, tau/amyloid/FDG-PET), cerebrospinal fluid (CSF) biomarkers and neuropathology [8, 10, 47–49]. Given the development of multicenter databases and computational techniques, integrating high-dimensional, multi-modality and longitudinal data is particularly appealing. That being said, regardless of the modality and clustering approach, it is crucial to recognize possible mechanisms underlying AD heterogeneity. APOE ε4 genotype, age-related and dementia-specific brain pathology, among others, have been proposed as putative drivers [9]. Cerebrovascular burden, as measured with the white matter hyperintensities Fazekas scale, were similar across two clusters in our study. We were unable to examine the effect of genetic data and tau/amyloid profiles on cluster membership, but previous researchers have suggested that atypical AD patients had lower probability of APOE ε4 and less severe neurofibrillary tangle (tau) pathology, which may have an impact on tau-targeting therapies in AD [7].

## NMF among other data-driven clustering approaches to unravel AD complexity

In neuropsychological research, the most commonly used clustering algorithms include hierarchical clustering, optimization clustering (including k-means clustering) and model-based clustering [50]. With the advances of computational methods in recent years, more diverse approaches have been employed by researchers. NMF is a data-driven clustering method that has been implemented to understand heterogeneity in AD in various studies [51–54]. In the

field of computational biology, its applications ranged from molecular pattern discovery, class comparison and prediction, to biomedical informatics and others with success as it could unravel biologically meaningful clusters [20, 21, 24]. Its objective is to interpret high dimensional data with a reduced number of components, or matrices, that when combined correspond to the observed data as close as possible, rendering it similar in concept to principal component analysis (PCA) and other dimensionality reduction techniques [19]. A distinctive feature of NMF is that it imposes a nonnegativity constraint on the factorized matrices, making it more complex algorithmically. Despite this, NMF remains popular due to its simple explanation of the factors, in contrast with the negative-sign results from PCA that often contradict physical reality and lack intuitive interpretation [21].

## Limitations

Our study is limited by the relatively small sample size and lack of pathologic confirmation of AD diagnosis. Several variables were not available for all patients, for example duration of onset and MRI measurements, as data were pulled from hospital-based outpatient records with some patients only undergoing brain CT, or having MRI scans from other hospitals that were not harmonized into our center's database. Furthermore, lack of data on comorbidities, genetic profiles, CSF and/or plasma biomarkers also precluded us from investigating their possible role as drivers of the underlying heterogeneity.

Finally, without evidence of AD postmortem neuropathology, as is the case with most studies in clinical settings, we cannot rule out the possibility that the results could have been partly driven by misdiagnosis, i.e. dementia due to cerebrovascular disease, frontotemporal lobar degeneration, or Lewy body disease [55]. For instance, in comparison with AD patients, those with vascular dementia demonstrate better performance on verbal learning and worse on executive functions, while scores in tests of language, attention, and visuospatial abilites were similar between the two groups [56]. Importantly, each type of dementia is a highly heterogeneous entity by itself, hence evaluation based on neuropsychological test profiles is challenging due to individuals with different dementias showing overlapping patterns of cognitive impairment. In this study, we attempted to include participants with AD based on careful selection from a cohort of dementia patients being followed up at our medical institution. As AD dementia remains primarily a clinical diagnosis and the sensitivity and specificity of the NIA-AA diagnostic criteria have been shown to be 71%, we believe that the likelihood of misdiagnosis remains a concern but has been minimized as best as we could [55]. Future efforts will concentrate on acquiring the missing data from a larger cohort of patients, as well as validating the result of clustering technique across time and subsets of patients with available neuropathological examination.

## Conclusion

Our results confirmed that patients with mild to moderate AD demonstrate cognitive heterogeneity as two subgroups with predominant deficits in memory and non-memory domains, respectively. Whether these cognitive profiles have prognostic implications, i.e. show different rates of progression, remains inconclusive. Large-scale, longitudinal and multicenter cohorts with well-phenotyped AD participants will strengthen the findings of cognitive heterogeneity and elucidate its causal mechanisms.

## Supporting information

**S1 Table. Cognitive Abilities Screening Instrument (CASI) version C–2.0.**
(DOCX)

**S1 Fig. Cophenetic correlation coefficients for r ranging from 2 to 6.**
(TIFF)

## Acknowledgments

We wish to express our gratitude to the patients whose data were included for analysis in this study.

## Author Contributions

**Conceptualization:** Truc Tran Thanh Nguyen, Hsun-Hua Lee, Wei-Chung Vivian Yang, Ming-Chin Lin.

**Data curation:** Truc Tran Thanh Nguyen, Li-Kai Huang, Chaur-Jong Hu, Chih-Yang Yeh.

**Formal analysis:** Truc Tran Thanh Nguyen.

**Funding acquisition:** Wei-Chung Vivian Yang, Ming-Chin Lin.

**Investigation:** Truc Tran Thanh Nguyen, Wei-Chung Vivian Yang, Ming-Chin Lin.

**Methodology:** Truc Tran Thanh Nguyen, Hsun-Hua Lee, Chih-Yang Yeh.

**Project administration:** Chih-Yang Yeh.

**Resources:** Chih-Yang Yeh, Ming-Chin Lin.

**Software:** Truc Tran Thanh Nguyen.

**Supervision:** Chaur-Jong Hu, Wei-Chung Vivian Yang, Ming-Chin Lin.

**Visualization:** Truc Tran Thanh Nguyen.

**Writing – original draft:** Truc Tran Thanh Nguyen, Hsun-Hua Lee, Li-Kai Huang, Chaur-Jong Hu, Ming-Chin Lin.

**Writing – review & editing:** Truc Tran Thanh Nguyen, Chaur-Jong Hu, Wei-Chung Vivian Yang, Ming-Chin Lin.

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
