## [Decision Letter · Decision Letter 0]

5 Jul 2023

PONE-D-23-03358Heterogeneity of Alzheimer's disease identified by neuropsychological test profilingPLOS ONE

Dear Dr. Lin,

Thank you for submitting your work to PLOS ONE and apologies for the protracted delay in the review process. Please make the corrections posed by Reviewer #1 so I can render a decision on this manuscript.

**Comments to the Author**

1. Is the manuscript technically sound, and do the data support the conclusions?

Reviewer #1: Yes

2. Has the statistical analysis been performed appropriately and rigorously? 

Reviewer #1: Yes

3. Have the authors made all data underlying the findings in their manuscript fully available?

Reviewer #1: No

4. Is the manuscript presented in an intelligible fashion and written in standard English?

Reviewer #1: Yes

5. Review Comments to the Author

Reviewer #1: REVIEWER’S REPORT

Heterogeneity of Alzheimer's disease identified by neuropsychological test profiling

In the submitted manuscript, the authors aimed to investigate cognitive heterogeneity in Alzheimer’s Disease (AD), one of the open questions in the literature on dementia and its cognitive correlates.

To this end, they enrolled a sample of patients (N=112) with a clinical diagnosis of mild-moderate AD. All patients were subjected to a neurocognitive assessment (MMSE, CASI, CDR) and a subset of them also underwent a neuroimaging acquisition session (either CT or MRI).

The authors applied a data-driven approach (nonnegative matrix factorization) to identify cognitive clusters in their sample of patients and were able to identify two clusters characterized by predominant memory (62.5%) vs. non-memory (37.5%) deficits. Patients in the memory group had also a worse cognitive performance at baseline.

Overall, the paper is well-written, has several methodological strengths (analytic methods are also well described), and deals with a critical issue, namely the heterogeneity of cognitive profiles in AD patients and the need to map such heterogeneity to promote individualized treatments.

The following are some minor points that hopefully could contribute to improving the present manuscript before its publication.

- The introduction (and the discussion) would benefit from a wider discussion of literature on clusters in Alzheimer’s disease. Indeed, there is a bunch of studies showing different numbers of clusters, despite some overlap with the results of the present study (e.g., Zangrossi et al., 2021, doi: 10.3233/JAD-210719, Journal of Alzheimer’s Disease; Cappa et al., 2014, Behav Neurol). I believe that a discussion of the present results in the light of previous studies showing different (but somehow matching) results would contribute to the strength and robustness of the present paper

- Neuroimaging-derived parameters are poorly discussed. They are not entered in the NMF clusterization and in subsequent analyses. Why did not the authors include such measurements in the linear mixed-effect model? If this is not a variable of interest I suggest to clearly state it in the introduction or in the “clinical measurements” paragraph.

- Baseline MMSE: please, discuss why the clustering could not just be interpreted as representing different degrees of disease severity (since there are differences in baseline MMSE).

- Longitudinal trajectory: please clarify whether patients were followed for 3 (line 241) or 5 years (line 231) and which data were included (e.g., only the first 3 years of follow-up). Moreover, did authors take into account the time distance between timepoints (e.g., see Tondo et al., 2021, Env Res and Pub Health)

- Since biomarkers are not employed in this study, I think tthe authors should discuss more in depth in the limitation section whether the results could simply represent different pathologies (e.g., different dementias)

6. PLOS authors have the option to publish the peer review history of their article (what does this mean?). If published, this will include your full peer review and any attached files.

**Do you want your identity to be public for this peer review?** For information about this choice, including consent withdrawal, please see our Privacy Policy.

Reviewer #1: No

We look forward to receiving your revised manuscript.

Kind regards,

Stephen D. Ginsberg, Ph.D.

Section Editor

PLOS ONE

Journal Requirements:

2. Please describe in your methods section how capacity to provide consent was determined for the participants in this study. Please also state whether your ethics committee or IRB approved this consent procedure. If you did not assess capacity to consent please briefly outline why this was not necessary in this case.

This research was funded by the Ministry of Science and Technology, Taiwan (grant number 108-2314-B-038-053-MY3) to TTTN, CY, and MC, and supported from Taipei Medical University, Taiwan (grant number 108-FRP-02) to HH, LK, and MC.

---

## [Author Response · Author response to Decision Letter 0]

11 Sep 2023

PONE-D-23-03358

Heterogeneity of Alzheimer's disease identified by neuropsychological test profiling

PLOS ONE

REVIEWER #1 (Comments to the Authors)

In the submitted manuscript, the authors aimed to investigate cognitive heterogeneity in Alzheimer’s Disease (AD), one of the open questions in the literature on dementia and its cognitive correlates.

To this end, they enrolled a sample of patients (N=112) with a clinical diagnosis of mild-moderate AD. All patients were subjected to a neurocognitive assessment (MMSE, CASI, CDR) and a subset of them also underwent a neuroimaging acquisition session (either CT or MRI).

The authors applied a data-driven approach (nonnegative matrix factorization) to identify cognitive clusters in their sample of patients and were able to identify two clusters characterized by predominant memory (62.5%) vs. non-memory (37.5%) deficits. Patients in the memory group had also a worse cognitive performance at baseline.

Overall, the paper is well-written, has several methodological strengths (analytic methods are also well described), and deals with a critical issue, namely the heterogeneity of cognitive profiles in AD patients and the need to map such heterogeneity to promote individualized treatments.

We are thankful to the reviewer for their very detailed feedback on the manuscript. Addressing their concerns has significantly improved the manuscript, and substantiated the validity of the results.

The following are some minor points that hopefully could contribute to improving the present manuscript before its publication.

- The introduction (and the discussion) would benefit from a wider discussion of literature on clusters in Alzheimer’s disease. Indeed, there is a bunch of studies showing different numbers of clusters, despite some overlap with the results of the present study (e.g., Zangrossi et al., 2021, doi: 10.3233/JAD-210719, Journal of Alzheimer’s Disease; Cappa et al., 2014, Behav Neurol). I believe that a discussion of the present results in the light of previous studies showing different (but somehow matching) results would contribute to the strength and robustness of the present paper.

R1.1. We thank the reviewer for their suggestion. We have expanded the Introduction and Discussion to include references that extended our study results. They read as follows: 

"For instance, Scheltens et al. employed a data-driven clustering method on neuropsychological measurements and identified two subgroups of AD with discrete cognitive profiles (Scheltens et al., 2017; Scheltens et al., 2018). One subgroup, the "memory" AD, had lower scores in memory-related tests, more severe hippocampal atrophy, slower disease progression and lower risk of mortality, compared to the remaining "non-memory" group that showed impairment in the language or executive/visuospatial domain. This finding strongly indicates that cognitive heterogeneity could predict individuals who are at greater risk of deterioration in dementia." (line 68-75)

"Previous work by Cappa et al., which employed one of the most comprehensive NP battery, found four hierarchical clustering-derived subtypes, namely the memory, visuospatial/perceptual, perceptual/calculation, and language from a cohort of patients with AD and posterior cortical atrophy (Cappa et al., 2014). Similarly, Zangrossi et al. identified four AD groups – typical, mild, visuospatial, and nonamnestic – that were partly explained by differences in sex and global cognitive functioning (Zangrossi et al., 2021). Scheltens et al. employed latent class analysis and identified eight AD cognitive clusters characterized by discrete neuropsychological performance and disease severity (Scheltens et al., 2016)." (line 361-368)

Scheltens NME, Tijms BM, Koene T, Barkhof F, Teunissen CE, Wolfsgruber S, et al. Cognitive subtypes of probable Alzheimer’s disease robustly identified in four cohorts. Alzheimer’s & Dementia. 2017;13(11):1226–36.

Scheltens NME, Tijms BM, Heymans MW, Rabinovici GD, Cohn-Sheehy BI, Miller BL, et al. Prominent Non-Memory Deficits in Alzheimer’s Disease Are Associated with Faster Disease Progression. J Alzheimers Dis. 2018;65(3):1029–39.

Cappa A, Ciccarelli N, Baldonero E, Martelli M, Silveri MC. Posterior AD-Type Pathology: Cognitive Subtypes Emerging from a Cluster Analysis. Behavioural Neurology. 2014:e259358.

Zangrossi A, Montemurro S, Altoè G, Mondini S. Heterogeneity and Factorial Structure in Alzheimer’s Disease: A Cognitive Perspective. J Alzheimers Dis. 2021;83(3):1341–51.

Scheltens NME, Galindo-Garre F, Pijnenburg YAL, Vlies AE van der, Smits LL, Koene T, et al. The identification of cognitive subtypes in Alzheimer’s disease dementia using latent class analysis. J Neurol Neurosurg Psychiatry. 2016;87(3):235–43.

- Neuroimaging-derived parameters are poorly discussed. They are not entered in the NMF clusterization and in subsequent analyses. Why did not the authors include such measurements in the linear mixed-effect model? If this is not a variable of interest I suggest to clearly state it in the introduction or in the “clinical measurements” paragraph.

R1.2. We thank the reviewer for their thorough reading. They are correct in that neuroimaging variables were not used extensively in this study (only in Table 1, where we compared the medial temporal lobe atrophy (MTA) score and white matter hyperintensities/Fazekas scale between the two subgroups). They were also not included in the linear mixed-effect model as they were only available at baseline examination of the participants, and not during follow-ups. 

 One alternative is to use neuroimaging software, such as FreeSurfer, to perform cortical parcellation and subcortical segmentation, which will provide a richer dataset in this study. Here we have opted to include the MTA and Fazekas measures, which are visual rating scales that can easily be applied in clinical settings. 

 We have added these points in the manuscript at line 123-127, 331-333, and 338-341.

- Baseline MMSE: please, discuss why the clustering could not just be interpreted as representing different degrees of disease severity (since there are differences in baseline MMSE).

R1.3. We thank the reviewer for this insight. We agree with the reviewer that heterogeneity can be caused by both disease subtype and disease stage. The field's opinions on this matter has been somewhat mixed. On the one hand, it could be interpreted that the variance identified from subtype analysis is actually driven by different disease stages, thus not true (cognitive) heterogeneity per se (Habes et al., 2020). On the other hand, one could argue that temporal and phenotypic heterogeneity are unavoidably intertwined and develop algorithms that take into account both simultaneously, as was done in Young et al., 2018. 

 In our study, we believe that disease stage did not substantially contribute to the cognitive subgroups, because the "age at disease onset" and "duration of onset" did not differ significantly between the two groups (Table 1). One more thing we could do is to validate the cognitive clusters after stratification for disease severity, i.e. split the cohort into two groups, mild and moderate in terms of severity, using the median MMSE/CASI/CDR scores, and repeat the cluster analysis on each group to see if the results would be similar (Scheltens et al., 2017). We elected to not perform this because the original sample size was only n = 112 and splitting the cohort further would likely result in too small sample size. Hence, we have added these arguments to the manuscript at line 316-329. 

Habes M, Grothe MJ, Tunc B, McMillan C, Wolk DA, Davatzikos C. Disentangling Heterogeneity in Alzheimer's Disease and Related Dementias Using Data-Driven Methods. Biol Psychiatry. 2020;88(1):70-82. 

Young AL, Marinescu RV, Oxtoby NP, et al. Uncovering the heterogeneity and temporal complexity of neurodegenerative diseases with Subtype and Stage Inference. Nat Commun. 2018;9(1):4273. 

Scheltens NME, Tijms BM, Koene T, et al. Cognitive subtypes of probable Alzheimer's disease robustly identified in four cohorts. Alzheimers Dement. 2017;13(11):1226-1236. 

- Longitudinal trajectory: please clarify whether patients were followed for 3 (line 241) or 5 years (line 231) and which data were included (e.g., only the first 3 years of follow-up). Moreover, did authors take into account the time distance between timepoints (e.g., see Tondo et al., 2021, Env Res and Pub Health).

R1.4. We sincerely thank the reviewer for their thorough reading and apologize for the confusion. 

 We would like to clarify that there were three instances where longitudinal data were involved in our study. Firstly, the total number of years of follow-up of each patient was collected and compared between those in the memory and non-memory group (Table 1). Secondly, we visualized the trajectory of disease progression via raw scores of MMSE, CASI, and CDR (Fig 3). Finally, we compared the rate of disease progression between two subgroups using linear mixed effect model analysis (Table 2 and Fig 4). 

 Below we show the number of participants that had available NP measures throughout the years in our study: 

TIME Baseline Year 1 (Y1) Y2 Y3 Y4 Y5 Y6 Y7 Y8

NUMBER OF PARTICIPANTS 112

(100%) 83

(74%) 76

(68%) 65

(58%) 43

(38%) 19

(17%) 7 

(6%) 3 

(2%) 1

(0.8%)

 For the linear mixed effect model analysis, we only included data within 3 years of follow-up because from the 4th year on the number of participants with available data dropped substantially (n = 43 from 112), which could induce the linear mixed effect model algorithm to fail to converge (i.e. it would not reach a stable solution with maximum likelihood, or participant with only 2 years of follow-up has more missing values than one that was followed for, say, 5 years). For the visualization of disease trajectory using raw scores, however, because there were no statistical analyses involved, we opted to include data of up to 5 years (Rongve et al., 2016). We have added the following sentence to clarify this in the manuscript, line 255-259.

"We only included data within three years of follow-up as from the fourth year on the number of participants with available data dropped substantially (n = 43 from 112), which could induce the linear mixed effect model algorithm to fail to converge, i.e. it would not reach a stable solution with maximum likelihood."

 Regarding whether we took into account the time distance between timepoints, we believe that our approach and that by Tondo et al., Env Res and Pub Health 2021 are different. Tondo et al. calculated the "index of progression" based on the formula follow-up MMSE−baseline MMSE/years of follow-up and compared this index between the three 2020, 2019, and 2018-groups using Kruskal–Wallis test. Hence, they were interested in only two timepoints (T0 and T1), which were mostly 1 year apart (e.g. "Follow-up in years" 0.93 ± 0.15 for the 2020 group), and excluded any patients with missing data. In contrast, our study included participants that were followed up for different number of years (e.g. the "memory" group were followed for 3.4 ± 1.8 years, range 0 - 8). The difference between subsequent timepoints is generally 1 year, give or take a few weeks to one month, depending on when the patient came in for re-evaluation, but we coded it as year 2, year 3, etc. for simplicity. For studies with missing data, longitudinal analysis with linear mixed effect model would be more advantageous than repeated measures ANOVA (Krueger C, Tian L, 2004). The latter can only handle missing data via use listwise deletion, which could reduce study power substantially. Finally, linear mixed effect model is also commonly used to compare disease progression between groups with different pattern of cognitive impairment (Scheltens et al., 2018; Qiu et al., 2019; Rongve et al., 2016). We therefore respectfully wish to maintain this section as it is. 

Rongve A, Soennesyn H, Skogseth R, et al. Cognitive decline in dementia with Lewy bodies: a 5-year prospective cohort study. BMJ Open. 2016;6(2):e010357.

Krueger C, Tian L. A comparison of the general linear mixed model and repeated measures ANOVA using a dataset with multiple missing data points. Biol Res Nurs. 2004;6(2):151-157.

Scheltens NME, Tijms BM, Heymans MW, et al. Prominent Non-Memory Deficits in Alzheimer's Disease Are Associated with Faster Disease Progression. J Alzheimers Dis. 2018;65(3):1029-1039.

Qiu Y, Jacobs DM, Messer K, Salmon DP, Feldman HH. Cognitive heterogeneity in probable Alzheimer disease: Clinical and neuropathologic features. Neurology. 2019;93(8):e778-e790. 

- Since biomarkers are not employed in this study, I think the authors should discuss more in depth in the limitation section whether the results could simply represent different pathologies (e.g., different dementias)

R1.5. We thank the reviewer for this suggestion. We have addressed this issue in the Discussion (Limitations, line 430-466), which reads as follows: 

"Finally, without evidence of AD postmortem neuropathology, as is the case with most studies in clinical settings, we cannot rule out the possibility that the results could have been partly driven by misdiagnosis, i.e. dementia due to cerebrovascular disease, frontotemporal lobar degeneration, or Lewy body disease (Beach et al., 2012). For instance, in comparison with AD patients, those with vascular dementia demonstrate better performance on verbal learning and worse on executive functions, while scores in tests of language, attention, and visuospatial abilites were similar between the two groups (Looi et al., 1999). Importantly, each type of dementia is a highly heterogeneous entity by itself, hence evaluation based on neuropsychological test profiles is challenging due to individuals with different dementias showing overlapping patterns of cognitive impairment. In this study, we attempted to include participants with AD based on careful selection from a cohort of dementia patients being followed up at our medical institution. As AD dementia remains primarily a clinical diagnosis and the sensitivity and specificity of the NIA-AA diagnostic criteria have been shown to be 71%, we believe that the likelihood of misdiagnosis remains a concern but has been minimized as best as we could (Beach et al., 2012). Future efforts will concentrate on acquiring the missing data from a larger cohort of patients, as well as validating the result of clustering technique across time and subsets of patients with available neuropathological examination."

Beach TG, Monsell SE, Phillips LE, Kukull W. Accuracy of the Clinical Diagnosis of Alzheimer Disease at National Institute on Aging Alzheimer Disease Centers, 2005–2010. Journal of Neuropathology & Experimental Neurology. 2012;71(4):266–73. 

Looi JC, Sachdev PS. Differentiation of vascular dementia from AD on neuropsychological tests. Neurology. 1999;53(4):670–8.

 

ACADEMIC EDITOR

Journal Requirements:

We have checked and ensured that the manuscript has met your style requirements.

2. Please describe in your methods section how capacity to provide consent was determined for the participants in this study. Please also state whether your ethics committee or IRB approved this consent procedure. If you did not assess capacity to consent please briefly outline why this was not necessary in this case.

Please see our answer to your 3rd requirement.

Thank you very much. We have included the following information in the manuscript: 

"This is a retrospective study of medical records, all data were fully anonymized before they were accessed, and the IRB committee waived the requirement for informed consent." (line 98-100)

This research was funded by the Ministry of Science and Technology, Taiwan (grant number 108-2314-B-038-053-MY3) to TTTN, CY, and MC, and supported from Taipei Medical University, Taiwan (grant number 108-FRP-02) to HH, LK, and MC.

Thank you very much. Please help us to include the Role of Funder statement as follows: 

The Joint Institutional Review Board of Human Research at the Taipei Medical University has restricted access to the de-identified data set used in this study. Data requests may be sent to Biomedical Technology Building, Shuang-Ho campus, No. 301, Yuantong Road, Zhonghe District, New Taipei City, ohr@tmu.edu.tw.

6. PLOS requires an ORCID iD for the corresponding author in Editorial Manager on papers submitted after December 6th, 2016. Please ensure that you have an ORCID iD and that it is validated in Editorial Manager. To do this, go to ‘Update my Information’ (in the upper left-hand corner of the main menu), and click on the Fetch/Validate link next to the ORCID field. This will take you to the ORCID site and allow you to create a new iD or authenticate a pre-existing iD in Editorial Manager. Please see the following video for instructions on linking an ORCID iD to your Editorial Manager account: https://www.youtube.com/watch?v=_xcclfuvtxQ.

Thank you very much. We have updated the ORCID iD of the study's corresponding author. 

Changes to the reference list have been mentioned in the rebuttal letter. We confirmed that no papers in our reference list have been retracted as of present.

Thank you very much. We have used PACE to ensure that the figures meet PLOS requirements as you requested.

---

## [Editor Report · Decision Letter 1]

25 Sep 2023

Heterogeneity of Alzheimer's disease identified by neuropsychological test profiling

PONE-D-23-03358R1

Dear Dr. Lin,

We’re pleased to inform you that your manuscript has been judged scientifically suitable for publication and will be formally accepted for publication once it meets all outstanding technical requirements.

Kind regards,

Stephen D. Ginsberg, Ph.D.

Section Editor

PLOS ONE

---

## [Editor Report · Acceptance letter]

28 Sep 2023

PONE-D-23-03358R1 

Heterogeneity of Alzheimer's disease identified by neuropsychological test profiling 

Dear Dr. Lin:

I'm pleased to inform you that your manuscript has been deemed suitable for publication in PLOS ONE. Congratulations! Your manuscript is now with our production department. 

Kind regards, 

on behalf of

Dr. Stephen D. Ginsberg 

Section Editor

PLOS ONE